# A pseudoproxy assessment of data assimilation for reconstructing the atmosphere-ocean dynamics of hydroclimate extremes

Nathan Steiger[1] and Jason Smerdon[1]

[1]Lamont-Doherty Earth Observatory, Columbia University, Palisades, New York, USA

*Correspondence to:* Nathan Steiger (nsteiger@ldeo.columbia.edu)

**Abstract.**

Because of the relatively brief observational record, the climate dynamics that drive multi-year to centennial hydroclimate variability are not adequately characterized and understood. Data assimilation(DA)-based paleoclimate reconstructions optimally fuse paleoclimate proxies with the dynamical constraints of climate models, thus providing a coherent dynamical picture of the past. DA is therefore an important new tool for elucidating the mechanisms of hydroclimate variability over the last several millennia. But DA has so far remained untested for global hydroclimate reconstructions. Here we explore whether or not DA can be used to skillfully reconstruct global hydroclimate variability along with the driving climate dynamics. Through a set of idealized pseudoproxy experiments we find that an established DA reconstruction approach can in principle be used to reconstruct hydroclimate at both annual and seasonal time scales. We find that the skill of such reconstructions is generally highest near the proxy sites. This set of reconstruction experiments are specifically designed to estimate a realistic upper-bound for the skill of this DA approach. Importantly, this experimental framework allows us to see where and for what variables the reconstruction approach may never achieve high skill. In particular for tree rings, we find that hydroclimate reconstructions depend critically on moisture-sensitive trees, while temperature reconstructions depend critically on temperature-sensitive trees. Real-world DA-based reconstructions will therefore likely require a spatial mixture of temperature and moisture sensitive trees to reconstruct both temperature and hydroclimate variables. Additionally, we illustrate how DA can be used to elucidate the dynamical mechanisms of drought with two examples: tropical drivers of multi-year droughts in the North American Southwest and in equatorial East Africa. This work thus provides a foundation for future DA-based hydroclimate reconstructions using real proxy networks, while also highlighting the utility of this important tool for hydroclimate research.

## 1   Introduction

Hydroclimate extremes, including persistent droughts and pluvials, can have extensive impacts on human welfare and agricultural production. While the frequency of such extremes are estimated to increase with global warming (e.g., Battisti and Naylor, 2009; Cook et al., 2015a; Ault et al., 2016), it is unclear if the underlying climate dynamics of such events are accurately produced in climate model simulations (e.g., Coats et al., 2016; Cook et al., 2016; Stevenson et al., 2016). The approximately 100-year observational record also limits our ability to understand the mechanisms of these events because this time interval does not capture the full range of hydroclimate extreme events and it poorly characterizes multidecadal to centennial variabil-

ity. A canonical example is the multidecadal "megadroughts" that were a prominent, though infrequent, feature of the climate system over the past two thousand years in Western North America and Europe (Cook et al., 2010b, 2015b, 2016). Of particular concern is that many of these drought events were more persistent than any of the droughts observed over the historical era (Cook et al., 2010b). Reconstructing hydroclimate variables along with their driving dynamics would therefore provide a critical perspective and allow for improved characterizations of such high-impact events.

While hydroclimate reconstructions exist over many different time scales and resolutions (e.g., Tierney et al., 2008, 2011; Masson-Delmotte et al., 2013), hydroclimate reconstructions over the Common Era (the past two thousand years) are particularly important because this time period allows for seasonal and annual time scale reconstructions over much of the globe (Bradley, 2014; Smerdon et al., 2017). Hydroclimate reconstructions for the Common Era have been performed for localized regions, such as for snowpack and precipitation in specific mountain ranges (e.g., Belmecheri et al., 2016; Neukom et al., 2015) or for the streamflow of particular rivers (e.g., Jacoby Jr, 1976; Cook et al., 2013). On continental scales, gridded "drought atlases" targeting the Palmer drought severity index (PDSI) have been derived over portions of North America (Cook et al., 2007; Stahle et al., 2016), Europe (Cook et al., 2015b), Southeastern Asia (Cook et al., 2010a), and Australia and New Zealand (Palmer et al., 2015). Despite these regional successes, no truly global hydroclimate reconstructions have been attempted.

The goal of this work is to demonstrate and test the application of data assimilation (DA) for global hydroclimate reconstructions. DA is a method that optimally fuses proxy information with the dynamical constraints of climate models (Goosse et al., 2012; Steiger et al., 2014; Hakim et al., 2016). We lay this groundwork by performing a set of synthetic "pseudoproxy" reconstructions that estimate a realistic upper-bound on reconstruction skill: given the current proxy networks and state-of-the-art climate models, we ask whether or not it is possible to skillfully reconstruct hydroclimate variables and their related dynamics. Pseudoproxy experiments provide a test of the reconstruction methodology while controlling for uncontrolled and sometimes unknown factors that pervade real-proxy reconstructions (see Smerdon, 2012, for a review). Pseudoproxy experiments have been successfully used to test the skill and characteristics of several reconstruction methodologies, including DA, and reconstructed variables such as temperature and precipitation (e.g., Mann et al., 2005, 2007; Christiansen et al., 2009; Smerdon et al., 2011; Werner et al., 2013; Steiger et al., 2014; Wang et al., 2014, 2015; Gómez-Navarro et al., 2015). The present pseudoproxy reconstruction work will help to focus future DA reconstruction efforts on variables, time scales, and geographic areas that are most likely to be successfully reconstructed. This paper also showcases the new kinds of analyses that are possible with the dynamical atmosphere-ocean variables from a DA-based climate reconstruction. We first analyze the pseudo-proxy reconstructions by focusing on global scale temperature and hydroclimate indices. We then move into specific analyses of the mechanisms of drought in the North American Southwest and equatorial East Africa.

## 2 Experimental Framework

### 2.1 DA Methodology

We employ a DA technique that optimally combines observations (in this context, proxy data) with climate model states. The model provides an initial, or prior, state estimate that is updated in a Bayesian sense based on the observations and an estimate

of the errors in both the observations and the prior. The prior may contain any climate model variables of interest. The updated prior, called the posterior, is the estimate of the climate state given the observations and the error estimates. The basic state update equations of DA (e.g., Kalnay, 2003, chapter 5) are given by

$$\mathbf{x}_a = \mathbf{x}_b + \mathbf{K}[\mathbf{y} - \mathcal{H}(\mathbf{x}_b)], \tag{1}$$

where $\mathbf{x}_b$ is the prior (or "background") estimate of the state vector and $\mathbf{x}_a$ is the posterior (or "analysis") state vector. Obser-
vations (or proxies) are contained in vector $\mathbf{y}$. The observations are estimated by the prior through $\mathcal{H}(\mathbf{x}_b)$, which is, in general, a nonlinear vector-valued observation operator that maps $\mathbf{x}_b$ from the state space to the observation space. The Kalman gain matrix $\mathbf{K}$ can be written as

$$\mathbf{K} = \mathbf{B}\mathbf{H}^{\mathbf{T}}[\mathbf{H}\mathbf{B}\mathbf{H}^{\mathbf{T}} + \mathbf{R}]^{-1}, \tag{2}$$

where $\mathbf{B}$ is the prior covariance matrix, $\mathbf{R}$ is the error covariance matrix for the proxy data, and $\mathbf{H}$ is the linearization of $\mathcal{H}$
about the mean value of the prior. Specifically, we use an ensemble square root filter (Whitaker and Hamill, 2002) to compute Eq. (1). We also note that Eq. (1) assumes that $\mathbf{y}$, $\mathbf{x}_b$, and $\mathcal{H}(\mathbf{x}_b)$ are Gaussian distributed and that their errors are unbiased. The reconstruction process works by essentially computing an optimal linear fit between the initial guess of the climate state (the prior $\mathbf{x}_b$) and the proxies ($\mathbf{y}$). This process involves iteratively computing Eq. (1) for each year of the reconstruction to arrive at ensemble estimates (the posterior $\mathbf{x}_a$) of the climate for each year. In any given year of the reconstruction, $\mathbf{x}_a$ is the
probability distribution of states that are consistent with the proxy observations and errors, as well as the prior distribution; therefore it is the full probability distribution of $\mathbf{x}_a$ that represents the probabilistic reconstruction of the climate state. In the time series figures herein, we show the mean of $\mathbf{x}_a$, with uncertainty estimates derived from the posterior distribution. For more mathematical details of the reconstruction methodology and the precise calculation procedure, see the Appendix of Steiger et al. (2014).
25       As in previous studies (Steiger et al., 2014; Hakim et al., 2016; Steiger and Hakim, 2016; Dee et al., 2016; Steiger et al., 2017) we use an "offline" DA approach in which the prior distribution is drawn from existing climate model simulations. For this approach, the ensemble members are seasonally or annually-averaged climate states instead of an ensemble of independently running model simulations, as in "online" DA. In principle, the ensemble members can be drawn from a single long simulation or multiple simulations or even from simulations of a collection of climate models; the only important requirement is that the
prior be climatologically representative of what one is trying to reconstruct (e.g., to reconstruct a year with a large volcanic eruption, the prior should contain ensemble members that come from simulation years with large volcanic eruptions). Because of how the prior is constructed here, it does not contain year-specific forcings or boundary conditions, which information appears to be superfluous according to many previous reconstructions experiments (Steiger et al., 2014; Hakim et al., 2016; Steiger and Hakim, 2016; Dee et al., 2016; Steiger et al., 2017; Okazaki and Yoshimura, 2017). The reconstruction process is also performed for each year independently such that no information is propagated forward in time. As discussed in Steiger

et al. (2014); Hakim et al. (2016); Steiger et al. (2017), the off-line approach is well-suited to a paleoclimate context where it has been shown to be highly skillful at annual time scales (Hakim et al., 2016) without the immense computational costs of a traditional online approach. Moreover, tests of offline vs. online approaches for paleoclimate have so far shown no improvement in reconstruction skill with an online method (Matsikaris et al., 2015; Acevedo et al., 2016).

## 2.2  Climate Model Simulations

Here we explore hydroclimate reconstructions using DA with a series of pseudoproxy experiments. For these experiments, we employ two full-forcing simulations of the Community Earth System Model from the Last Millennium Ensemble Project (CESM LME) (Otto-Bliesner et al., 2016). These CESM simulations used a $\sim$2-degree atmosphere and land, and $\sim$1-degree ocean and sea ice components. The simulations were run from the years 850 to 1850 CE using estimates of the transient evolution of solar intensity, volcanic emissions, greenhouse gases, aerosols, land-use conditions, and orbital parameters. The

simulations were given identical forcings but differed by round-off error in the initial atmospheric state; this difference was sufficient to generate simulations with different time histories (e.g., Nino 3.4 indices from the two simulations we use here have an annual average correlation of 0.097). These simulations provide the climate states for our prior ensembles as well as the climate inputs for the pseudoproxies: the pseudoproxies, $\mathbf{y}$, are generated for the CESM LME simulation 10 while the prior, $\mathbf{x}_b$, and the prior estimate of the pseudoproxies, $\mathcal{H}(\mathbf{x}_b)$, are from simulation 9. In other words, simulation 10 plays the

role of the actual climate in a real-world reconstruction setting; the pseudoproxies are all derived from that simulation, while simulation 9 generates the ensemble of states that define the prior. We use two different simulations in this way so that there is no overlap between the specific prior states and the reconstructed climate states.

## 2.3  Pseudoproxy Construction

We generate realistic pseudoproxies via process (or forward) models for each proxy type (see Evans et al., 2013, for a review),

which transform the simulated climate signal (e.g., temperature, precipitation) into synthetic proxy observations. For the pseudoproxy experiments, we construct pseudo tree-ring width and an approximation of pseudo-coral $\delta^{18}O$. We only consider these two proxy types because they have established proxy system models and because they represent the majority of proxies in global databases [e.g., $92\%$ of proxies in the PAGES2k network (PAGES2k Consortium, 2017)]. Because our experimental construction defines an upper-limit on reconstruction skill, we do not consider age uncertainties, which would act to degrade

reconstruction skill and are not applicable for the tree-ring proxies that constitute the majority of the proxy network. We use the pseudoproxy network shown in Fig. 1, which is based on the updated PAGES2k network of real, global proxy data (PAGES2k Consortium, 2017). In this pseudoproxy network, we created only one pseudoproxy per climate model grid cell, though in real proxy reconstructions there would be the possibility of having multiple proxies within a given grid cell. Reconstructions were performed using both pseudo-tree and pseudo-coral proxies together and also using pseudo-trees alone.

We generated 160 pseudo tree-ring widths using the model VS-lite (Tolwinski-Ward et al., 2011), which uses monthly temperature and precipitation from the CESM LME simulations to calculate tree-ring width time series at each pseudo tree

5  ring location. This model accounts for the seasonal dependence of tree growth via simple parameterizations of temperature

and moisture threshold responses at monthly scales, as well as latitudinal light availability. We estimated the four growth parameters of VS-lite for each pseudo-tree location using the real PAGES2k tree ring width data (PAGES2k Consortium, 2017), the monthly observational temperature and precipitation data of CRU TS3.23 (Harris et al., 2014), and the VS-lite parameter estimation code from Tolwinski-Ward et al. (2013). Note also that because this proxy system model takes care of the averaging time scale of each individual tree, there is no need to define a rigid time scale imposed on all tree sites (e.g., all trees in the Northern Hemisphere reflect only June-July-August temperature).

We generated 55 pseudo-coral "$\delta^{18}O$" through the linear proxy system model of Thompson et al. (2011), which for a given coral proxy can be written as

$$\delta^{18}O_{pseudocoral} = a_1 \text{SST} + a_2 \text{SSS} + \epsilon, \tag{3}$$

where SST is local, annually averaged sea surface temperature and SSS is local, annually averaged sea surface salinity. We estimated the parameters $a_1$ and $a_2$ based on the real coral proxies in the PAGES2k database and annual (defined as April to the next March) SST and SSS from the ocean reanalysis dataset SODA (Carton and Giese, 2008). Note that the CESM simulations are not isotope-enabled and the coral proxy system model only requires SST and SSS to create coral "$\delta^{18}O$" pseudoproxies.

Within the formulations for both the pseudo tree-ring widths and pseudo coral "$\delta^{18}O$," each have realistic parameters estimated with real proxy and historical data, but are driven off-line by climate time series inputs from the CESM LME simulations, viz., 2 m air temperature, total precipitation, SST, and SSS. While the combination of the climate simulation output and the proxy system models produce single proxy values for a given year, these values represent the many short time-scale processes included in the climate model simulations, such as daily weather events.

We additionally note a special issue regarding the use of the VS-lite proxy system model within our Kalman filter-based DA method: we find the output of VS-lite for some locations to be strictly non-Gaussian. This is important because Eq. (1) assumes that $\mathbf{y}$ and $\mathcal{H}(\mathbf{x}_b)$ are Gaussian. In preliminary work, we observed "filter divergence" (where the state estimate diverges unrealistically away from the true state) for small prior ensembles when using VS-lite pseudoproxies. We have not observed this effect with the proxy system models for corals or ice cores (Dee et al., 2016; Steiger et al., 2017). A common, though ad hoc way of dealing with filter divergence is to employ covariance localization that smooths out spurious long-range correlations, acts to boost the "effective" ensemble size, and tends to reduce non-Gaussian issues (Houtekamer and Mitchell, 1998). Using covariance localization is unattractive, however, because it is ad hoc and ensemble sizes can be very large in an offline DA approach. We therefore found that to avoid filter divergence we needed over 500 ensemble members in our prior. We did not find much sensitivity to the results with ensemble sizes between about 600 and 1000 members (e.g., maximum changes of 0.05 in the correlation skill of the global mean temperature reconstructions). We therefore performed the reconstructions with 1000 members, the maximum available years for the CESM LME simulations. This does not limit future real proxy reconstructions because of the many millennial-length climate model simulations available (Fernández-Donado et al., 2013; Masson-Delmotte et al., 2013), but it did require that we use the CESM LME in the present pseudoproxy experiments because it is the only high resolution, CMIP5/PMIP3 class climate model with multiple millennial-length simulations that are publicly available.

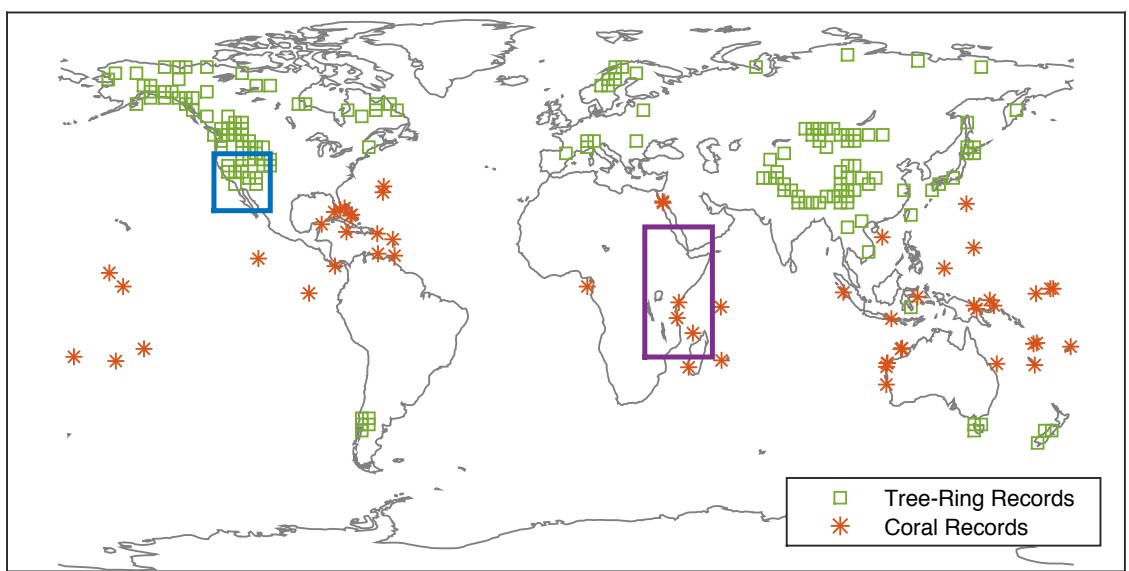

**Figure 1.** Pseudoproxy network used in this study, based on the PAGES2k global database of tree-ring and coral records (PAGES2k Consortium, 2017), and the drought regions explored in this study, the North American Southwest in blue and equatorial East Africa in purple.

A further idealization in these reconstructions is that we consider only measurement error. Specifically, we add white noise with a standard deviation of 0.01 (approximate width measurement uncertainty) to the pseudo-tree-ring time series that already have a variance equal to 1 and 0.1 (approximate isotope measurement uncertainty) to the pseudo-coral time series that are in units of per mil; the diagonal elements of $\mathbf{R}$ are $0.01^2$ for pseudo-trees and $0.1^2$ for pseudo-corals. This choice is motivated by our interest in the best-case scenario so as to estimate an upper-bound for DA-based reconstructions. Our choice is also motivated by the fact that quantifying errors in non-linear proxy system models like VS-lite is non-trivial (Evans et al., 2014), thus we can have more confidence in an upper-bound reconstruction skill estimate than a "most-realistic" skill estimate. However, in the context of traditional regression-based reconstruction approaches, multivariate signals in proxies are "noise." For reference, when the no-noise-added tree ring and coral pseudoproxies are compared against the climate model PDSI and SST respectively, the tree rings have a mean signal-to-noise ratio (SNR) of 0.46 with a standard deviation of 0.50 and the corals have a mean SNR of 0.41 with a standard deviation of 0.23 (calculated following Mann et al. (2007), SNR $= |\rho|/\sqrt{1-\rho^2}$, where $\rho$ is the correlation between the proxy and the nearest grid point climate variable time series.) These SNRs are therefore typical of estimates for real proxies and are comparable to the SNRs adopted in pseudoproxy contexts that test traditional regression approaches (Wang et al., 2014).

## 2.4 Reconstruction Procedure and Experiments

Here we employ a "perfect model" framework where the same underlying climate model is used for both the prior and the "true" climate model state. The prior, $\mathbf{x}_b$, consists of all the climate states at their relevant resolutions (including monthly, seasonal,

and annual resolutions as defined below) from the CESM LME simulation number 9. Using this prior, we reconstruct the corresponding annual and seasonal states of simulation 10 based only on our prior, and the pseudoproxies that are generated from simulation 10. Despite the idealization of the perfect model framework, previous DA-based reconstruction work has shown similar results using different climate models in both pseudoproxy and real proxy reconstructions that did not assume a perfect model framework (Hakim et al., 2016; Dee et al., 2016; Steiger et al., 2017). These studies saw similar basin and

continent-wide patterns and magnitudes of reconstruction skill.

For the reconstructions, the annual states are constructed out of monthly CESM LME output, averaged from April to the next calendar year's March. This annual average is chosen to account for the seasonal cycle of a global network of proxies as well as climate phenomena like the El Niño–Southern Oscillation (ENSO), the continuity of which would be ignored with a calendar year average. The seasonal states we use are June-July-August (JJA) and December-January-February (DJF) averages

of the monthly climate variables.

We include the following global variables in our state vectors: 2 m air temperature, the Palmer drought severity index (PDSI), the standardized precipitation evapotranspiration index (SPEI) using a 12 month decaying exponential weighting kernel (Beguería et al., 2014) chosen to closely resemble the time scale of PDSI. Both PDSI and SPEI were computed using the Penman-Monteith equation for potential evapotranspiration and monthly climate model output of precipitation, 2 m temper-

ature, vapor pressure, net surface radiation, surface pressure, and surface wind (estimated from 10 m down to 2 m using the wind profile power law). Here we focus on PDSI and SPEI because they are so widely used for current drought monitoring and historical drought reconstructions. In addition to these global variables, we also include the following index variables: the location of the inter tropical convergence zone (ITCZ) defined by the precipitation centroid (center of mass) of the zonal mean tropical precipitation ($20°$N to $20°$S) (Frierson and Hwang, 2012), and the monthly Nino 3.4 index. Note that both of

these variables are computed upfront and included as index variables in the prior state vector rather than being post-processed from reconstructed spatial climate fields (both approaches give equivalent results); this was done simply to save computational memory and speed up the reconstructions.

As in previous work (Hakim et al., 2016; Steiger and Hakim, 2016; Dee et al., 2016; Steiger et al., 2017), we quantify uncertainty in our reconstructions through a combination of Monte Carlo sampling of the proxy network and accounting for

the spread in the posterior distribution, $\mathbf{x}_a$. Each 1000 year long reconstruction is repeated 25 times in a Monte Carlo fashion by sampling $75\%$ of the pseudoproxy network for each reconstruction iteration (we only perform 25 iterations because the small pseudoproxy errors lead to very similar reconstructions). This gives us 25 equally-likely reconstructions that each come with a 1000-member posterior ensemble estimate for each year of the reconstruction. Because this generates very large data files, we save the ensemble mean for each year of the Monte Carlo iterations and the full posterior ensembles for only one

of the iterations. This yields two distributions for each year based on the proxy network sampling and the spread in climate model states. We compute the standard deviation of each of these distributions and add them in quadrature, giving us the final uncertainty estimates for all of the variables in space and time. For this particular experimental setup, we note that it is actually only necessary to save one full posterior ensemble because the posterior spread depends only on the spread in the prior, the spread in the prior estimate of the proxies, the proxy error estimates, and the number of proxies [see, for example, Eqs. A2-A4

in Steiger et al. (2014)], each of which are identical for all the Monte Carlo iterations. The upshot is that the ensemble spread (but not the ensemble mean) is identical for each year and each reconstruction iteration.

All of the reconstruction experiments here have been designed to show upper-bound skill estimates. This experimental framework allows us to highlight what is theoretically possible while also making falsifiable claims about what is impossible to reconstruct. To summarize, experimental idealizations discussed previously include the following: a "perfect model" setup

where the covariance structures of $\mathbf{B}$ are derived from the same climate model (but not the same simulation) as the truth and $\mathcal{H}$ is known perfectly; low experimental error where $\mathbf{R}$ represents only measurement error, is based on white noise, and is assumed to be diagonal (proxy errors are uncorrelated). In a real reconstruction, the climate model covariances will not be identical to the covariance structures of Earth's climate, the proxy system model $\mathcal{H}$ will be imperfectly known, and proxies will include "errors" that do not reflect climate variability and are larger than measurement errors (Evans et al., 2013). Using this DA

approach, previous studies have looked at the effects of each of these idealizations within a pseudoproxy framework (Steiger et al., 2014; Dee et al., 2016; Steiger et al., 2017). These studies show that the uncertainties related to the imperfect climate models, proxy system models, and proxies will likely act to degrade reconstruction skill but not to the point where DA-based reconstructions provide no meaningful information. However, because such experiments are synthetically constructed, there is significant uncertainty about their fidelity to the real climate reconstruction problem; for example, Dee et al. (2016) used a

model of intermediate complexity to reconstruct a simulation from a state-of-the-art coupled model to test the effect of inexact model covariance structures of the prior, but it is unclear how representative this comparison is to a real-world reconstruction. Working within a more idealized context circumvents such issues but also affects the framing and interpretation of the results. These upper-bound skill estimates indicate what is theoretically possible and are most useful in showing which regions and variables lack skill and are thus likely unreconstructable.

## 3    Global Reconstruction Skill Assessment

We first assess the spatial temperature reconstructions in Fig. 2, which shows the skill of the reconstruction at each grid point using the metrics of correlation (r) and the mean continuous ranked probability skill score (CRPSS). Seasonal (JJA and DJF) and annual reconstructions are organized by column. Correlation is computed using only the reconstruction mean time series at each grid point while the CRPSS metric accounts for both the mean grid point time series as well as the grid point uncertainty

estimates. CRPSS is based on the continuous ranked probability score (CRPS), which is a "strictly proper" scoring rule that accounts for the skill of the entire posterior reconstruction distribution (Gneiting and Raftery, 2007); because the posterior ensemble estimates are normally distributed, we use Eq. (5) from Gneiting et al. (2005), which is given by

$$\text{crps} = \sigma \left\{ y_n \left[ 2\Phi(y_n) - 1 \right] + 2\phi(y_n) - \frac{1}{\sqrt{\pi}} \right\}, \tag{4}$$

where $y_n = (y - \mu)/\sigma$, with $y$ being the true value, $\mu$ the mean of the posterior ensemble estimate, and $\sigma$ the standard deviation of the posterior ensemble, and where $\phi(y_n)$ and $\Phi(y_n)$ are respectively the normal probability density function and the normal cumulative distribution function of $y_n$. All of our uses of Eq. (4) are for time series, either individual time series or grid point

5    time series. We therefore compute the mean of all the time-step values and denote it as CRPS. The skill score version, CRPSS, is the reconstructed CRPS computed with respect to the CRPS of a reference distribution, $\mathrm{CRPSS} \equiv 1 - \mathrm{CRPS_{rec}}/\mathrm{CRPS_{ref}}$, here the initial uninformed prior; positive CRPSS indicates that the reconstructed distribution is more skillful for this metric than the uninformed prior. CRPSS is a much more stringent skill metric than correlation, so we focus most of our attention on CRPSS.

10    Returning again to Fig. 2 and focusing on the CRPSS results, we note that skill is highest in the densest proxy regions. Comparing the first and second columns in Fig. 2, skill is also much higher near the pseudo-trees during their primary growing season (note that the full growing season for each pseudo-tree site is not preset, but determined by VS-lite based on the local temperature and moisture conditions that allow or suppress growth). Despite these seasonal dependencies, the annual mean temperature reconstructions in the third column of Fig. 2 are skillful (positive CRPSS) despite the fact that the pseudo-trees are not responsive to the full annual cycle of climate. This is possible because the DA algorithm can exploit climate field covariance information across time-scales: see Steiger and Hakim (2016) and note that, for example, ENSO and the approximately 500 day thermal time scale of the global ocean mixed layer (Lintner and Neelin, 2008) are physical features of the climate system that can retain information across seasonal and annual time scales. The skill of the temperature reconstructions in Fig. 2 is high

5    across the tropics and is not solely dependent on the presence of corals there, though they do improve tropical reconstructions (see section 4.1).

One more subtle feature of Fig. 2 is that not all proxy locations show similar local temperature reconstruction skill, particularly for the pseudo tree rings. This suggests that certain pseudoproxy locations provide more local information about temperature than other locations. Importantly, the tree ring proxy system model VS-lite explicitly models how much of a given

10    site's growth is affected by either temperature or moisture limitations (the two growth controlling factors modeled by VS-lite). Following Tolwinski-Ward et al. (2013), we used VS-lite's growth responses (optional outputs of VS-lite) to calculate which locations were either moisture or temperature limited, expressed here by a "moisture limitation fraction," as shown in Fig. 3. Specifically, we computed the fraction of summer months over the entire simulation in which the growth response to soil moisture was less than the growth response to temperature; summer is defined as JJA in the Northern Hemisphere and DJF in the Southern Hemisphere. Values of this fraction approaching zero indicate a temperature-limited site, while values approaching one indicate a moisture-limited site.

Comparison of moisture limitation fractions with the reconstruction skill estimates in Fig. 2 shows that the locations with the highest CRPSS skill values correspond to locations that are most temperature limited. For instance, the four tree-ring sites

5    in Arctic Russia reveal a range of temperature or moisture limited responses. The CRPSS regional values in JJA follow these sensitivities, namely the most temperature-limited proxy generates the highest regional skill, while the eastern most proxy that is moisture-limited is associated with the lowest regional skill. This generally holds for other isolated locations or clusters of similarly-limited trees, such as for the cluster of temperature-limited sites in Fennoscandia or the cluster of moisture-limited sites in the Pamir mountain range. Note that the identification of certain moisture or temperature limited sites is due primarily

10    to the CESM model simulation output, so the identifications here may not correspond perfectly to the related real-world proxy sites.

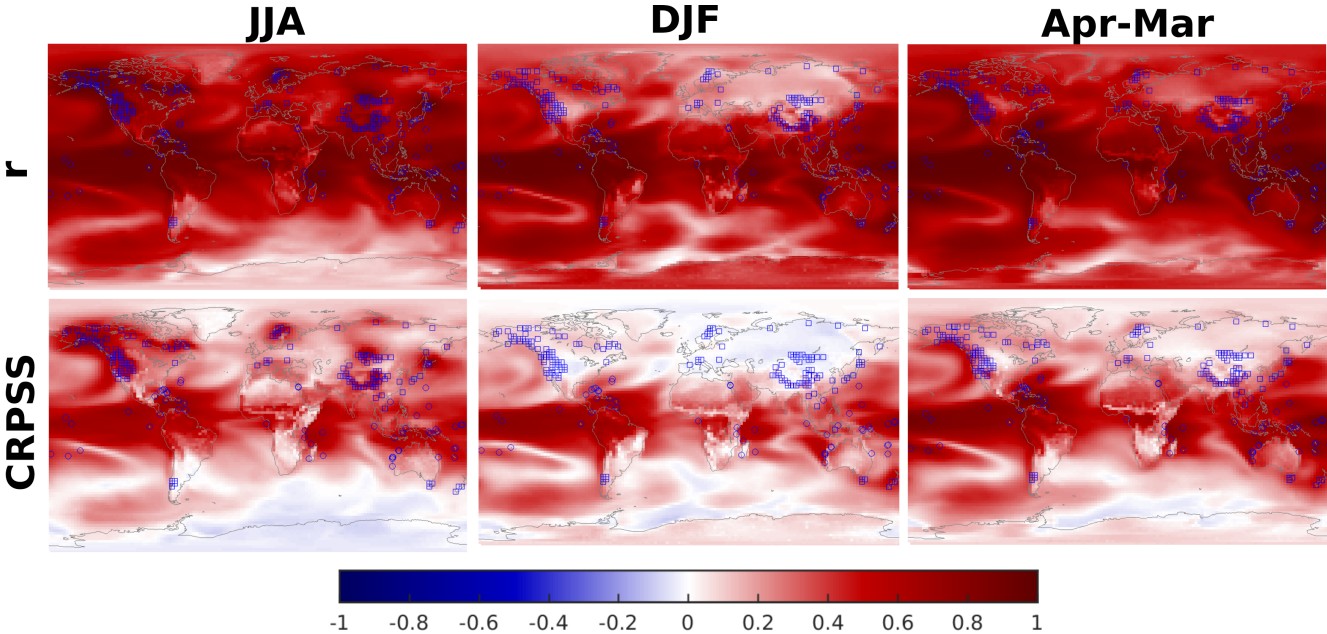

**Figure 2.** Reconstruction skill for 2 m air temperature. Rows show the skill metrics of correlation (r) and the mean continuous ranked probability skill score (CRPSS). These skill metrics are computed for the entire 1000 year reconstruction against the true grid point time series, CESM LME simulation 10. Columns show the reconstruction skill for June-July-August (JJA) and December-January-February (DJF) seasonal means, as well as the annual mean, April to the next calendar year's March. Pseudo-tree sites are indicated by blue squares, pseudo-corals by blue circles.

Figures 4 and 5 show the skill of the reconstructed PDSI and SPEI. Both indices behave very similarly, with slightly higher skill values for SPEI (e.g., over boreal Canada and Russia for JJA correlation and CRPSS). Similar to the temperature reconstruction, the higher skill values tend to be local to the proxy sites, especially for the CRPSS metric. This result particularly holds for the areas of single moisture-limited proxies or clusters thereof. These dependencies are again illustrated by comparing Figs. 3, 4, and 5 around the four Arctic Russia sites where skill is increased around the moisture-limited locations, particularly for the far eastern proxy in that region. Similarly, the cluster of moisture-limited sites in the Pamir mountain range is associated with increased skill there and westward to the Caspian sea. In contrast, the arc of Himalayan pseudo tree rings are largely temperature-limited and despite their density they have low CRPSS skill values for the PDSI and SPEI reconstructions. We also note that the skill over Australia and the equatorial regions is not dependent on the presence of the pseudo-corals (not shown), consistent with some of the findings of real proxy drought reconstructions (Palmer et al., 2015) that indicate the long covariance length scales in these regions. Similar to the temperature reconstructions, Figs. 4 and 5 show that skillful annual reconstructions of hydroclimate are in principle possible in this DA framework despite the fact that the pseudo-trees are primarily responding to growing season conditions. This result may be due in part to the persistence built into the drought indices.

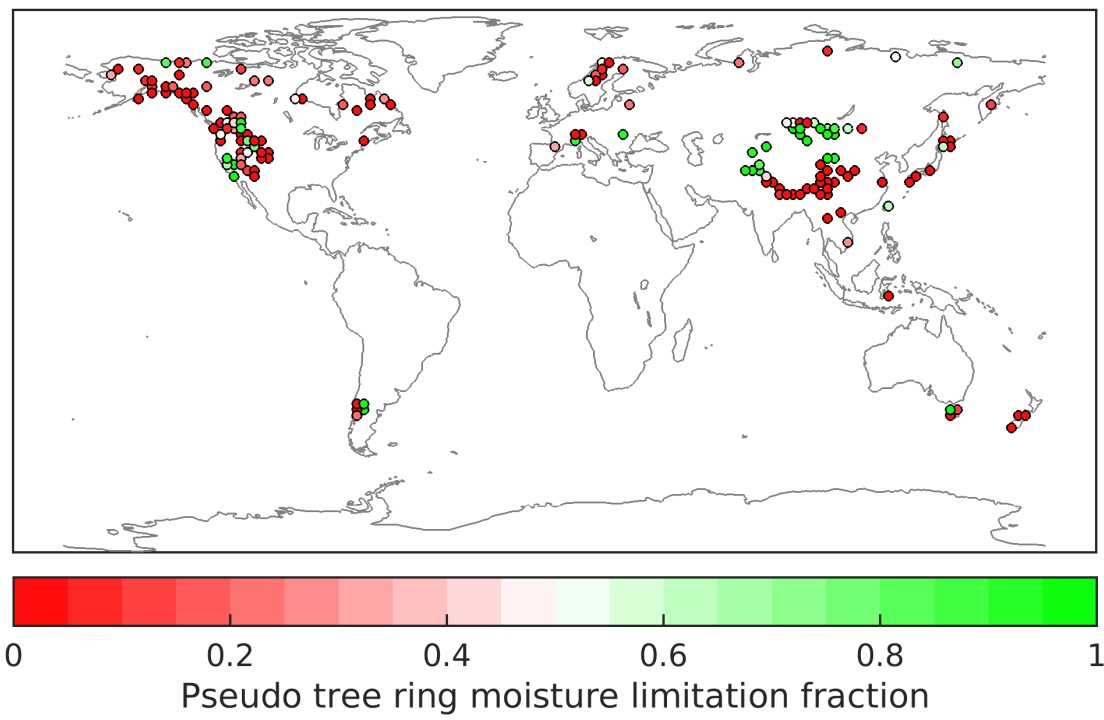

**Figure 3.** Moisture limitation fraction for each pseudo tree ring, with green values indicating more moisture-limited sites and red values indicating more temperature-limited sites. A value of 0 indicates a pseudo tree site that is completely temperature limited, while a value of 1 indicates a completely moisture-limited site.

Because the reconstructions shown here provide an upper-bound skill estimate, we expect Figs. 2, 4, and 5 to be most useful in showing where and why the reconstructions *lack* skill. If skill is limited under idealized conditions, it is unlikely that increased skill would be achieved in real reconstructions. In this case CRPSS is more useful because correlation is less stringent in that it only tracks the phase of the mean reconstruction and resulting in generally quite high correlations for most locations and seasons. The comparisons we have so far made between Fig. 3 and the skill metrics in Figs. 2, 4, and 5 illustrates a general rule that without moisture sensitive trees skillful PDSI or SPEI reconstructions are unlikely; similarly, without temperature sensitive trees skillful temperature reconstructions are unlikely. This might at first appear to be an obvious observation. It is nevertheless important to recall that PDSI and SPEI are both dependent on temperature through their incorporation of potential evapotranspiration. Therefore it might be possible in principle for tree rings to reconstruct both temperature and moisture indices equally well, regardless of whether the trees are moisture or temperature limited. To the contrary, our experiments suggest that this is not the case. Based on these experimental results, we recommend that future DA-based hydroclimate reconstructions use a spatial mixture of temperature and moisture sensitive trees if the goal is to reconstruct both temperature and hydroclimate indices (as happens to be the case over western North America in our experiments). Where this is impossible, our pseudoproxy

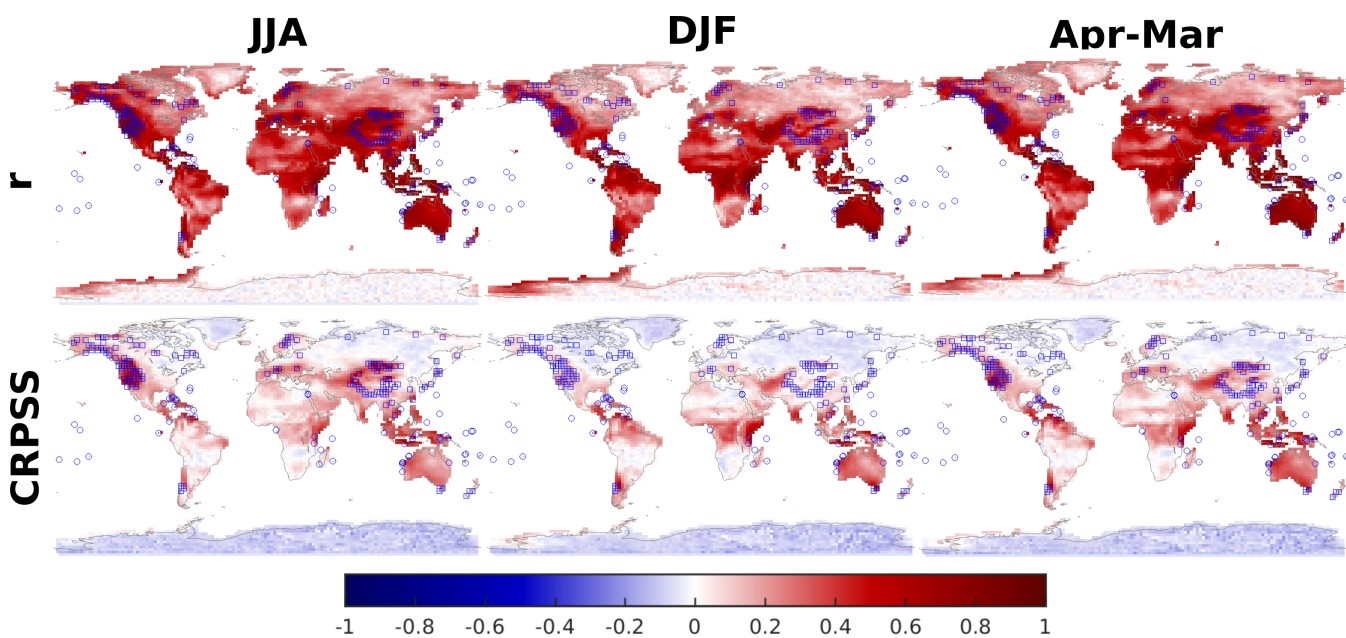

**Figure 4.** Reconstruction skill, correlation (r) and the mean continuous ranked probability skill score (CRPSS), for the Palmer drought severity index (PDSI), c.f. Fig. 2.

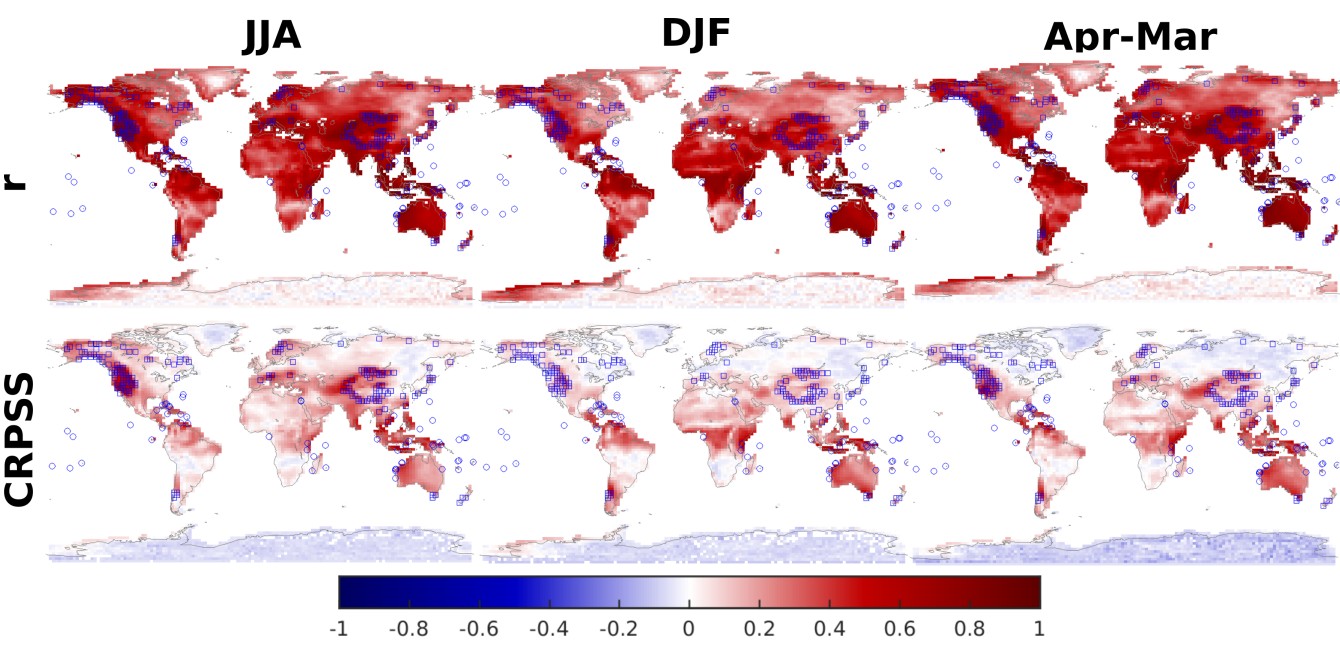

**Figure 5.** Reconstruction skill, correlation (r) and the mean continuous ranked probability skill score (CRPSS), for the standardized precipitation evapotranspiration index (SPEI) with a 12 month decaying exponential weighting kernel, c.f. Fig. 2 and 4.

experiments suggest that reconstruction skill will be significantly diminished for the target variable not represented by the proxy network.

## 4 Regional Hydroclimate Reconstructions

Given the above demonstrations of global reconstruction skill, we now showcase two regional examples that explore the driving dynamics of multi-year droughts in the North American Southwest and equatorial East Africa within the pseudoproxy reconstructions.

### 4.1 North American Southwest Multi-year Drought

We first focus on the dynamical mechanisms of multi-year droughts in the North American Southwest. The choice of this region is motivated by the local reconstruction skill seen in Figs. 4 and 5, and because several established theories exist about the dynamical drivers of drought in this region (e.g., Seager et al., 2005; Seager and Hoerling, 2014). Fig. 6 shows the time series reconstruction of JJA PDSI in the North American Southwest region, defined by the land area within the latitudinal range of 25°N–42.5°N and the longitudinal range of 125°W–105° W as previously used in Coats et al. (2013, 2015) and indicated by the blue box in Fig. 1. In Fig. 6, the five most severe droughts are highlighted for both the reconstruction and the truth; overlap is indicated by the slightly darker purple highlights. The cumulative drought severity is ranked by the sum of persistently negative PDSI values relative to an 11-year moving average (Meehl and Hu, 2006) [very similar results were found using an alternative definition of drought that commences after two consecutive years of negative PDSI values and continues until two consecutive years of positive PDSI values (Coats et al., 2013)]. Consistent with the spatial drought reconstructions shown in Figs. 4 and 5, this North American Southwest regional average reconstruction has high skill, r = 0.90 and CRPSS = 0.57 (we note that for a single time series verification CRPSS reduces to the skill score version of the mean absolute error, which we have computed here); the reconstruction using pseudo-trees alone has skill values of r = 0.89 and CRPSS = 0.54 (not shown). Despite the high skill of the overall reconstruction, it is challenging to consistently identify the rank of the most severe droughts based on the imperfect reconstruction; this is true for not only the top 5 droughts, but the top 10 and 15 as well. This is due to the fact that drought identification and ranking procedures (Meehl and Hu, 2006; Coats et al., 2013) tend to be sensitive to the precise zero crossings of the hydroclimate time series that set the beginning or ending of a drought (thus affecting both the identification and subsequent ranking); for example, the mid to late 1300s era drought in Fig. 6 is cut short in the reconstruction because of a zero crossing of the PDSI values near the middle of the true drought.

Given the established teleconnection between Pacific SST anomalies and hydroclimate variability in North America (Seager and Hoerling, 2014), we reconstruct the Nino 3.4 index and assess its relationship to North American Southwest multi-year droughts in CESM. As part of the reconstruction experiments, we reconstructed *monthly* Nino 3.4 with both pseudo-trees and pseudo-corals and as a separate reconstruction just with pseudo-trees. Fig. 7 shows the power spectra of the true Nino 3.4 time series along with the two reconstruction types. As a representative example of the reconstruction, Fig. 8 shows the monthly Nino 3.4 time series during the most severe drought during the reconstruction (this short interval was shown for clarity). Skill

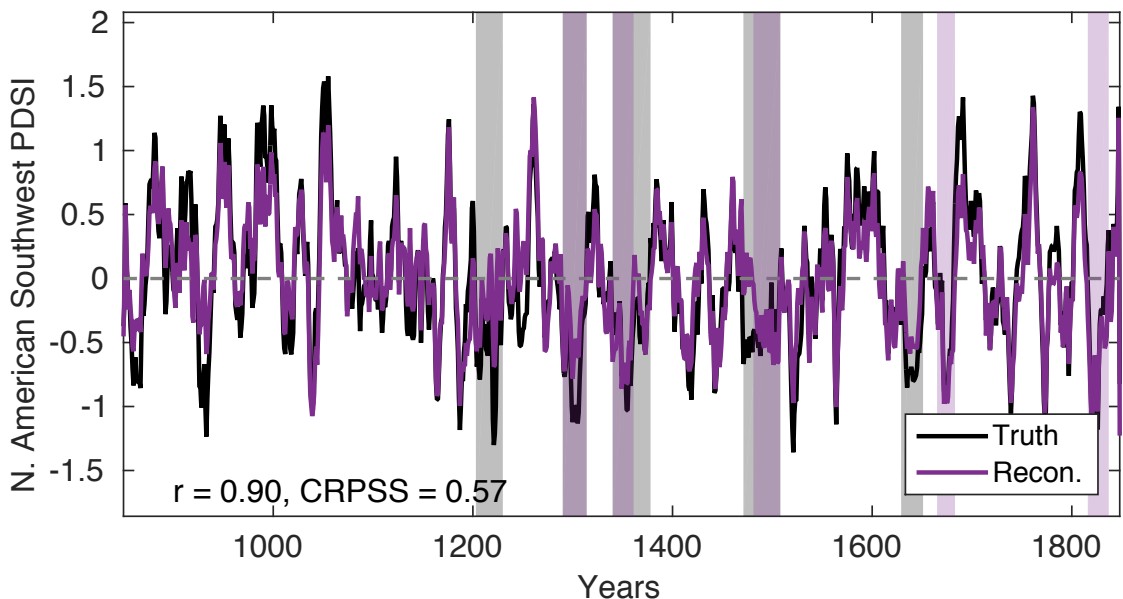

**Figure 6.** North American Southwest area-averaged JJA PDSI in the reconstruction mean (purple) and the climate model truth (black). Both time series have been smoothed with an 11-year moving average (Meehl and Hu, 2006). The skill metrics of correlation (r) and the mean continuous ranked probability skill score (CRPSS) for the smoothed time series are shown in the lower left corner. The five most severe multi-year droughts over this period are highlighted in corresponding colors. Reconstruction error estimates, similar to those shown in Fig. 8, have been left off to clearly show the severe drought highlighting.

scores for the entire monthly time series for the experiment assimilating both pseudo-trees and pseudo-corals were r = 0.97, CRPSS = 0.78, while for the experiment with pseudo-trees alone the skill values were r = 0.87, CRPSS = 0.53. Taking both Fig. 7 and Fig. 8 together, the pseudoproxy framework indicates that it is possible to skillfully reconstruct the monthly Nino

3.4 index, and to potentially do so without coral proxies. The monthly skill associated with the Nino 3.4 reconstruction is likely because monthly values sufficiently covary with the annual proxy observations such that the reconstruction can provide meaningful monthly states for this variable (Tardif et al., 2014; Steiger and Hakim, 2016). The results of the reconstructions using pseudo-trees alone is also particularly important given that coral records are generally short and not extant over more than a couple hundred years.

As an analysis of the relationship between North American Southwest drought variability and ENSO, Fig. 9 shows the monthly percent occurrence of El Niño or La Niña states during the 15 most severe multi-year droughts (spanning droughts of 9 to 38 years in length). The El Niño and La Niña states are defined as the months for which the SST anomaly exceeds $0.5°C$ and $−0.5°C$, respectively. Figure 9(a) shows how the percent occurrence for each drought reconstruction compares to the true percent occurrence, while Fig. 9(b) shows the distribution of all the 15 droughts for the truth and the two reconstructions. This

figure illustrates that in the CESM LME simulation, multi-year droughts are associated with an above average occurrence of La Niña states. This relationship is consistent with observations and historical model simulations, which in brief show that cooler

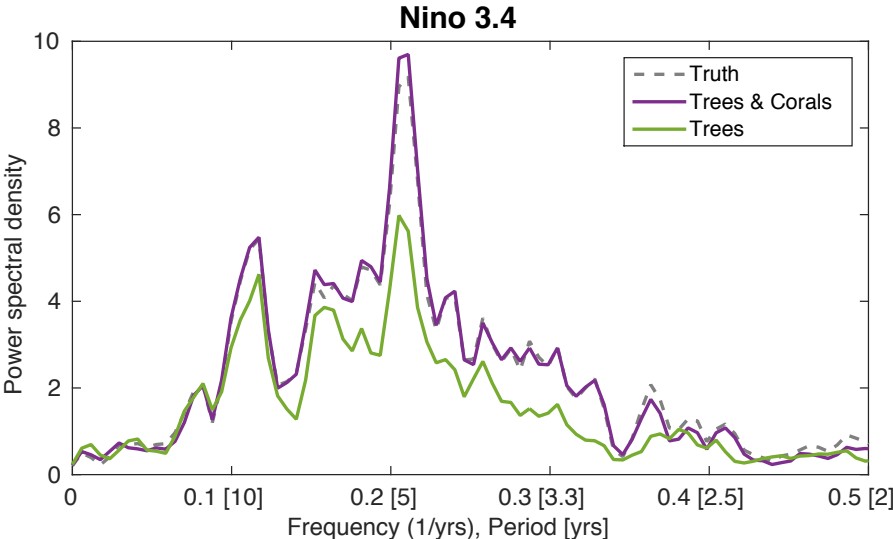

**Figure 7.** Power spectral density of the true model Nino 3.4 index time series along with the reconstructions based on pseudo-trees and pseudo-corals (purple) as well as pseudo-trees alone (green).

La Niña-like SSTs in the tropical Pacific tends to shift the storm track over North America polewards leading to a drying of the American Southwest (Seager and Hoerling, 2014). In a paleoclimate context, it has also been inferred that increased incidences of La Niña events have likely driven megadroughts in Western North America over the last 1,000 years (Coats et al., 2016).

Additionally, last millennium simulations of the CESM model have also been shown to have the strongest relationship between ENSO and drought in the American Southwest compared to other CMIP5-class models (Coats et al., 2015, 2016). These results thus collectively indicate that drought reconstructions and specific driving dynamics can be successfully investigated within in a DA-based reconstruction.

## 4.2  Equatorial East Africa Multi-year Drought

The next reconstruction illustration we consider is multi-year drought in equatorial East Africa. Specifically, we explore where drought is occurring and its plausible large-scale drivers according to the CESM LME simulations. Similar to the results shown in section 4.1, we use drought dynamics in East Africa as an example of what can be reconstructed using the DA-method. We additionally provide a dynamical analysis that would be the starting point for a region specific analysis that could be performed based on a real-proxy reconstruction. The choice of this region is motivated by the observation that the spatial drought index reconstructions, Figs. 4 and 5, show skillful drought reconstructions in Eastern Africa and because tropical dynamics and variables are generally better reconstructed than mid-latitude variables and dynamics in global, DA-based reconstructions [see, for example, Fig. 2 and Steiger et al. (2014); Hakim et al. (2016)].

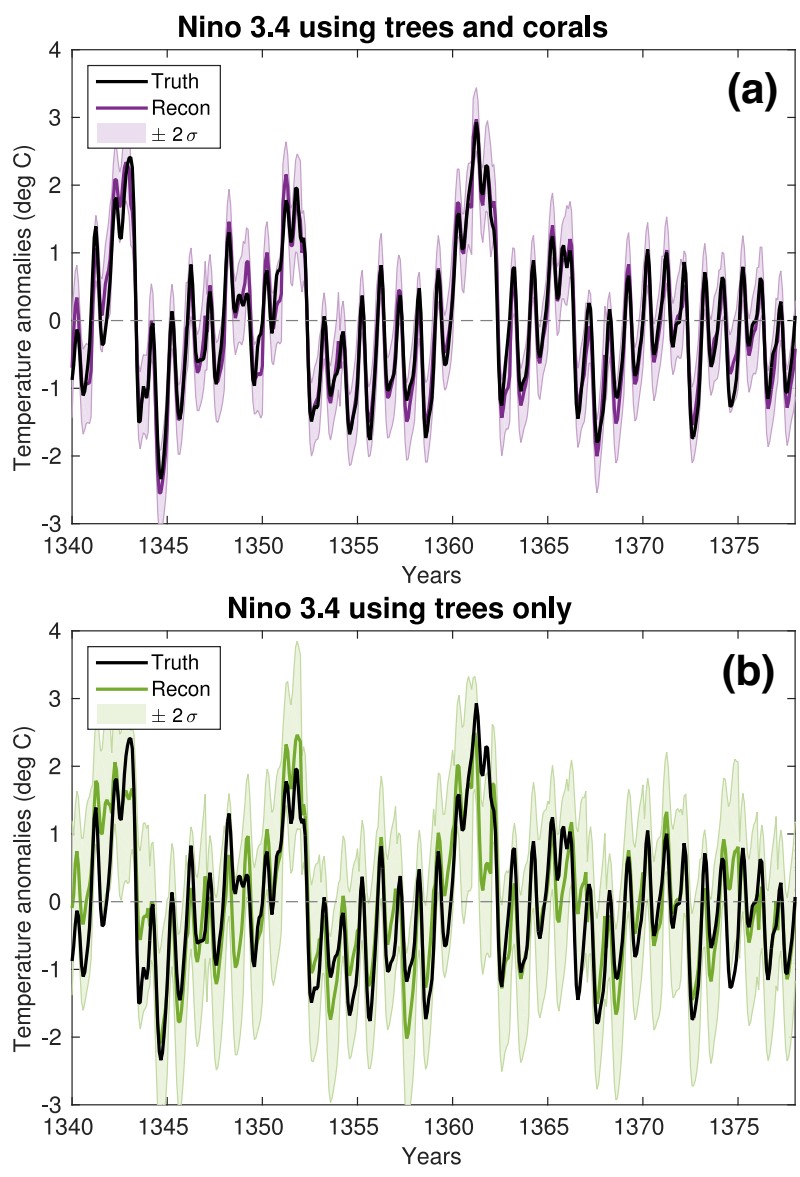

**Figure 8.** Monthly reconstruction of Nino 3.4 index during the most severe drought period in the reconstruction, for **(a)** using both pseudo-trees and pseudo-corals and **(b)** for using only pseudo-trees. The $2\sigma$ uncertainty estimates are derived from both the Monte Carlo reconstruction iterations and the posterior distribution, see section 2.4 for details.

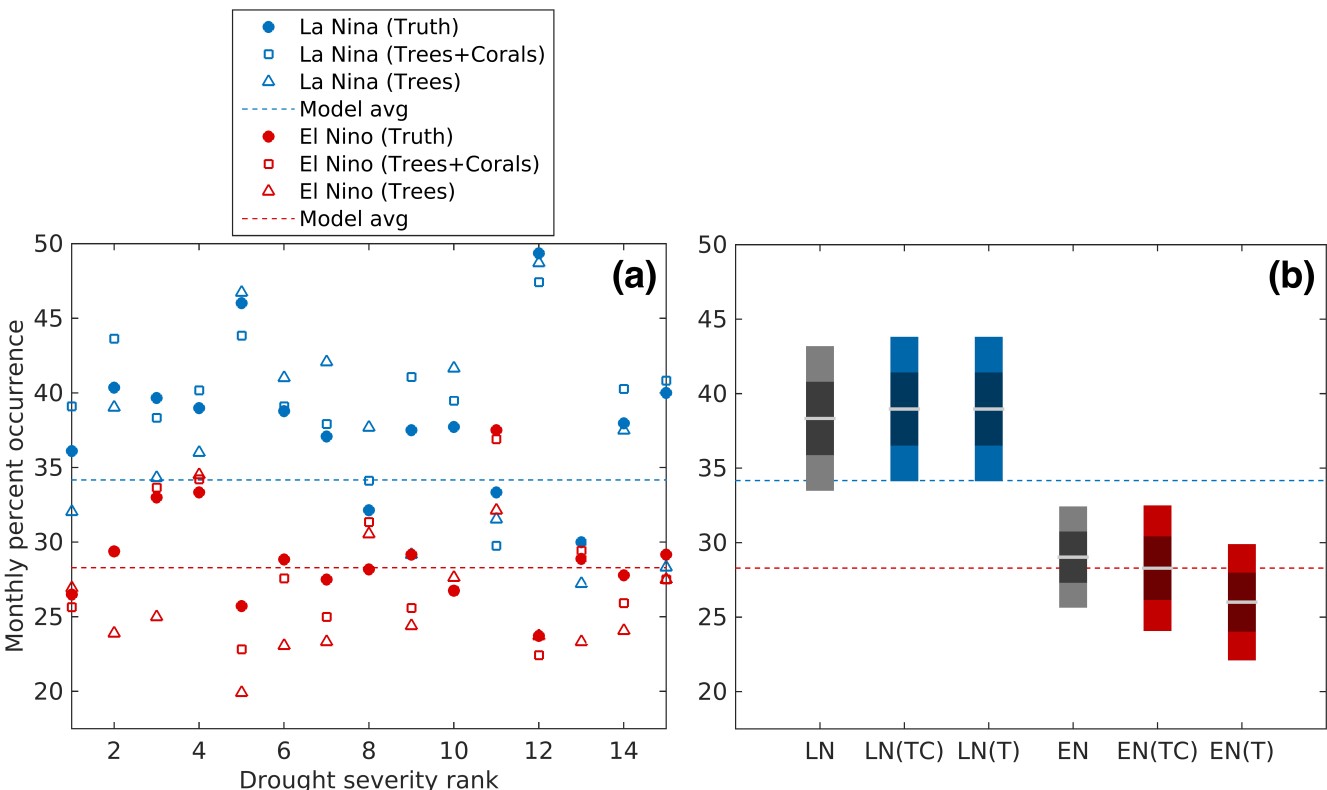

**Figure 9. (a)** Monthly percent occurrence of El Niño or La Niña states during the 15 most severe multi-year droughts. The occurrences are computed as a percentage of the total number of months in a given drought. Dashed lines indicate the average percent occurrence of El Niño (red) or La Niña (blue) in the CESM LME simulation 10. **(b)** Box plot distribution summaries of the monthly percent occurrence of El Niño or La Niña states shown in (a). Light gray lines indicate the means, the darker colors indicate one standard deviation of the data while the lighter colors indicate the 95% confidence intervals of the mean. The letters LN and EN stand for La Niña and El Niño, while the letters T and C indicate experiments with pseudo-trees or pseudo-corals. "LN" and "EN" by themselves are the true La Niña and El Niño occurrence (gray box plots), corresponding to the distribution of filled circles in (a).

For this analysis we assess how shifts in East African tropical circulation relate to multi-year droughts. An extensive number of paleoclimate studies have broadly interpreted tropical or even extratropical hydrologic changes to shifts in the ITCZ (see Chiang and Friedman, 2012, for a review). Here we explore how a common indicator of the location of the ITCZ, the zonal precipitation centroid (Frierson and Hwang, 2012), is related the location of East African droughts. We calculate the precipitation centroid as the center of mass of the zonal mean precipitation within the latitudinal range of 20°N–20°S and the longitudinal range of 28°E–52°E; as seen in Fig. 1 this boxed area is centered over Eastern Africa and includes precipitation over both land and ocean. We construct a similar drought location index by computing the centroid of the zonal mean negative SPEI values over the continental African land area within this East African box. We use SPEI for this region because some work indicates that it may be better suited to drought assessments in East Africa than PDSI (Ntale and Gan, 2003), though the results shown in this section are not dependent on the choice of drought variable.

Figure 10(a) shows the mean latitudinal locations of the 15 most severe multi-year droughts in equatorial East Africa for both the reconstruction and the model truth. Additionally, this figure shows the mean latitudinal location of the precipitation centroid averaged over each of the drought periods. The locations of the two centroids are well-matched to the truth in the reconstruction. Furthermore, Fig. 10(b) shows that these two centroids are negatively correlated (in both the truth and the reconstructions), indicating that as the average precipitation shifts south in CESM, drought occurs in the north and vice versa. Because the zonal mean precipitation is strongly peaked in the annual mean with more variability in the wings of the zonal mean distribution, small changes in the precipitation centroid can lead to larger changes in the location of drought (Fig. 10a). Similar to the results presented for North America, these collective results for East Africa further illustrate the utility of DA-based reconstructions and their potential to test specific hypotheses, such as the hypothesis that shifts in the ITCZ can drive multi-year droughts.

## 5   Conclusions

Using DA for hydroclimate reconstructions is a new area of research. The aim of this paper was therefore to explore some foundational aspects of global hydroclimate reconstructions using DA. Through a set of pseudoproxy reconstruction experiments, we estimated a realistic upper-bound on hydroclimate reconstruction skill for a global pseudoproxy network based on pseudo tree rings and corals. Unlike most pseudoproxy experiments that use statistical noise perturbations for proxy construction, the pseudo tree rings and corals were here generated using established proxy system models. These models are physically-based and account for multiple climatic factors on proxy development.

Because these reconstruction experiments have been designed as upper-bound skill estimates, the results are most useful in showing where and for which variables the reconstructions *lack* skill. This framing allows us to highlight what is theoretically possible while also making falsifiable claims about what is impossible to reconstruct given current proxy networks. In our experiments we find that skill is highest in the tropics and in regions local to proxy sampling, in accordance with many previous pseudo and real proxy reconstructions that focused on temperature or other non-hydroclimate variables (e.g., Smerdon, 2012; Steiger et al., 2014; Hakim et al., 2016; Anchukaitis et al., 2017). For tree rings (which constitute the majority of annually-

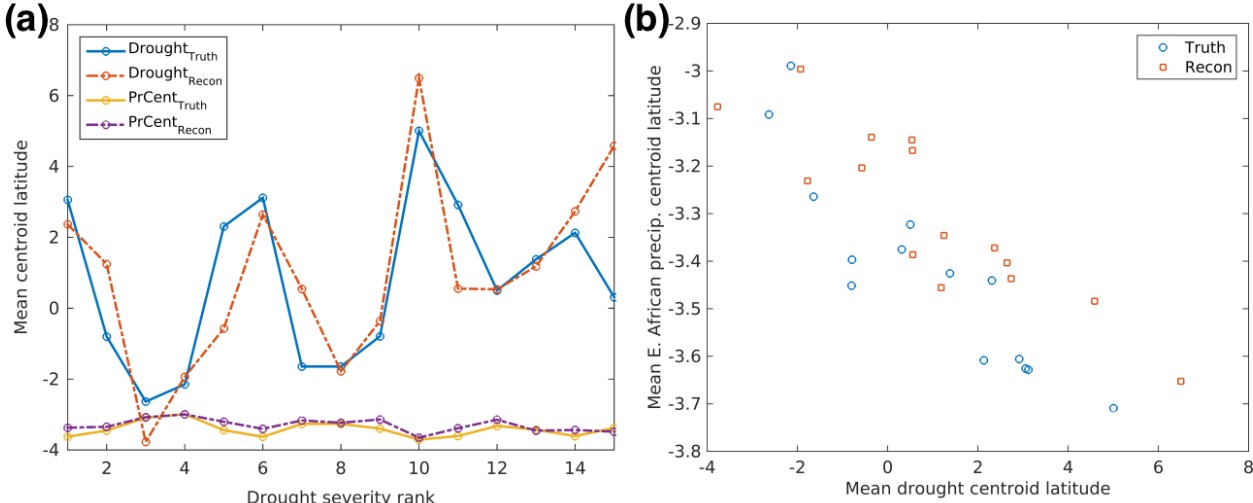

**Figure 10.** The latitude locations of the zonal mean multi-year drought and precipitation centroids (PrCent), for both the truth and the reconstruction over equatorial East Africa. Panel **(a)** shows the centroid locations as a function of drought severity rank, while panel **(b)** has the same information but more clearly shows the negative correlation between the locations of drought and precipitation centroids.

resolved proxies) we also find that local reconstruction skill depends on the moisture or temperature sensitivity of particular tree ring proxies: moisture sensitive trees are necessary for skillful PDSI or SPEI reconstructions and similarly, temperature sensitive trees are necessary for skillful temperature reconstructions. The covariability of moisture and temperature is not sufficiently strong for our DA approach to reconstruct both variables equally well from a given tree. Based on these results, we recommend that future DA-based hydroclimate reconstructions use a spatial mixture of temperature and moisture sensitive trees if the goal is to reconstruct both temperature and hydroclimate indices. Importantly, we also performed seasonal and monthly reconstructions and found that it is in principle possible to reconstruct non-growing season or annual mean climate variables in many regions using DA. We also found that it is possible to reconstruct a monthly Nino 3.4 index.

Two regional climate dynamics analyses also explored the key drivers of hydroclimate extremes in the North American Southwest and equatorial East Africa. These regional examples highlight how DA-based reconstructions could in general be used to find the dynamical drivers of hydroclimate extremes. This information-added benefit from DA-based reconstructions is very important given that dynamical information is difficult to simultaneously obtain from a traditional reconstruction. Based on these results, the application of DA to the PAGES2k network may yield skillful hydroclimate reconstructions that will deepen our current understanding of decadal to centennial hydroclimate variability. We find that this approach can potentially yield dynamical insights about regions that have not been previously well characterized, such as Africa.

## 6 Code availability

The code used in the production and analysis of the reconstructions is available upon request from nsteiger@ldeo.columbia.edu.

## 7 Data availability

Monthly CESM LME data is available from http://www.earthsystemgrid.org. The reconstructions from this paper will be made
publicly available through the Lamont Ingrid Data Library if the paper progresses to publication.

*Author contributions.* N.S. and J.S. designed the experiments and prepared the manuscript. N.S. developed the code, ran the experiments, and analyzed the experimental results.

*Competing interests.* The authors declare no competing interests.

*Acknowledgements.* We acknowledge the CESM1 (CAM5) Last Millennium Ensemble Community Project and the supercomputing re-
sources provided by NSF/CISL/Yellowstone. This work was supported by the NOAA Climate and Global Change Postdoctoral Fellowship Program, administered by UCAR's Visiting Scientist Programs. This work was also supported in part by the National Science Foundation, grants AGS-1243204, AGS-1401400, and AGS-1602581. LDEO contribution number XXXX.

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
