# Peer review of "A pseudoproxy assessment of data assimilation for reconstructing the atmosphere-ocean dynamics of hydroclimate extremes"

_Climate of the Past, 2017_

## Referee Comment (RC1) · Anonymous Referee #1 · 21 Jun 2017

This paper uses established data assimilation techniques to reconstruct not only temperature but also hydroclimate variability. It carries out a perfect model study where both the prior information and target climate come from the same model (CESM) and employs a pseudo-proxy technique to carry out as close to a real–world based reconstruction as possible. This allows a validation of the method because by design the truth is known in advance. The paper looks at the ability of the technique to reconstruct temperature and drought globally as well as regional multi-year drought and ENSO. I find the paper well written and a good first step to what I am sure will be further studies

that use this technique to make useful reconstructions of the real world climate.

My only main concern is that I do not think that the limitations of this perfect model study are made sufficiently clear. While in several places you mention that this is an "upper-bound on hydroclimate reconstruction skill" I think that this should be emphasised further, particularly in the abstract, but also while discussing the results. And it should be made clear that based on this study on its own you cannot tell if a skilful DA reconstruction is actually a possibility now, merely that it is theoretically possible. To this end I think that a section describing what uncertainty this technique includes and what it does not include would be very useful. For example the uncertainty in the pseudo-proxy parameter estimation is not included and because only one model is used model uncertainty is also not included. Given that there is model dependence in the moisture/temperature limitation sites (line 14 p 9) could this not be a real problem for this technique. To be clear, I have no problem with the results and techniques currently in the paper, and do not think that more work is needed, I merely think that slightly more detail is required to frame them in a more instructive way.

Apart from this I have only a few relatively minor comments (organised in roughly the order they appear in the text)

I think that title should be changed from "hydroclimate extremes" to "hydroclimate variability" to reflect the more general approach of the paper.

I found the description of the Kalman filter slightly hard to follow and think it could be improved by giving more information about what the filter is actually doing, in more accessible language so that a non-specialised audience can follow.

Do the matrices have a time dimension, or is the solution for each time step completely independent? This should be clarified.

Could multiple CESM ensembles be used for the prior information? Or does it have to be one continuous simulation?

[Figure]

More information should also be given about the uncertainty matrix R. Has it only diagonal terms of 0.1? Where does this come from and what uncertainty does it include and does it not include?

Figure 2 and others – On my print out it is quite hard to make out the position of the proxy sites in some panels.

Figure 9 - What percentiles are shown in the box plots?

---

## Referee Comment (RC2) · Anonymous Referee #2 · 2 Aug 2017

The paper reports on a pseudoproxy study designed to assess the possibility to reconstruct hydroclimatic variability using a data assimilation approach. The approach as well as several other aspects are described elsewhere (they use CESM simulations as background and proxies as observations - in the pseudoproxy study they use forward models to generate pseudoproxies), but here the focus is specifically on hydroclimatic variables.

The results are scientifically sound and the paper is well written. It is interesting for the community. However, the paper clearly gives a very optimistic upper bound. While

this is certainly important, it might be more useful to the reader to give a more realistic view. Where does the method break down? What are the limitations? It is hard to judge for the reader, particularly since several optimistic assumptions are only implicit and a discussion of the limitations is missing. While I don't think that the authors should add further analyses, I definitely think they should (1) be more clear about the assumptions and (2) add a discussion on the potential limitations when applying it to the real world.

Specifically, the methods assumes that there is no model bias, that the model B matrix represents the true covariance structure. It is not clear how the residuals for the pseudoproxy are calculated and whether H is assumed to be known perfectly (or are the parameters of H degenerated, or is the VS-lite recalibrated somehow with the pseudoproxies?). Is R taken as diagonal? What noise model is used? And is R assumed to be known perfectly?

These assumptions are very optimistic (even more optimistic is the low assumed error), and it is hard to judge how important it is. Teleconnections are assumed to be perfectly represented and linear and they are assumed to be stationary (constant in time, e.g., independent of forcings).

Another worry I have, in the examples given, is that the NINO index is specified with the same tree ring width series as the drought indices. Hence, the NINO index and drought are expected to be related as they are specified from the same tree ring width. Actually, this is seen in Fig. 9, which shows a stronger ENSO signal in the reconstructions using only tree ring width than in the "true" simulation. There is an element of circular reasoning here. In this context, localisation should be discussed. The author speak of localisation as an "ad hoc" method, but controlling this sort of circular reasoning would be one advantage that the authors should consider.

Minor

P. 3, l. 10: Give a reference for the Ensemble Square Root Filter implementation.

P. 3, l. 29: I think the boundary conditions require some further explanations. Is it sensible not to consider boundary conditions? Or would physical consistency be violated (e.g. by using a non-volcanic background during a volcanic year?

P. 6, l. 11: The amount of noise is really small; I am surprised by that.

P. 6-7: From the statement that a monthly NINO index was reconstructed I take that $x_b$ and $x_a$ contain both monthly and seasonal variables. How about the annual ones? Are annually and seasonal variables in the same state vector? Or are these two different experiments?

P. 7, l. 23: If possible within reasonable length, give equation and references for CRPSS.

P. 9, l. 4: It is not fully clear how the limitations were derived. As I understand the approach, the actual limitations are dependent on the climate conditions (couldn't the same VS-lite parameters make a tree moisture sensitive in one year but temperature sensitive in another one?).

Fig. 2 is interesting. In the upper row (which is not really discussed in the text), the high correlations over Antarctica are striking (this could be relevant for other reconstructions). Also in the second row I find the very high skill in the tropics remarkable (the authors note it, but I think it requires more explanation). Also, it is interesting that the skill in the annual mean is smaller over the proxy sites than over the adjacent oceans. This is due to winter, where there is no skill over the proxy locations but (due to thermal inertia or other memory effects) some limited skill over the ocean. Also, in the annual mean there seem to be prominent patterns (dipoles?) in the N- and S-Pacific, pointing to very stable teleconnections within the model world.

P. 12, l. 27: I presume that the assimilation is the same as above - or not?

P. 14, l. 3: In addition to El Niño, a look at the Atlantic Ocean might be interesting.

P. 14, l. 12-14: I find this conclusion a rather dangerous one to make in a perfect-model

set-up.

---

## Author Comment (AC1) · 22 Aug 2017

Response to Reviewers

We would like to thank both reviewers for helpful and constructive comments.

Besides minor textual and methodological clarifications that we will address point-by-point in a revised manuscript and response, the primary concern raised by both reviewers is the idealized nature of the experiments presented in the paper. For example, Reviewer 1 recommends (similar to comments from Reviewer 2):

*While in several places you mention that this is an "upper-bound on hydroclimate reconstruction skill" I think that this should be emphasised further, particularly in the abstract, but also while discussing the results. And it should be made clear that based on this study on its own you cannot tell if a skilfull DA reconstruction is actually a possibility now, merely that it is theoretically possible. To this end I think that a section describing what uncertainty this technique includes and what it does not include would be very useful.*

This is a good idea and we will add further clarification on this point in the abstract of the revised manuscript and in new a summarizing paragraph in the Conclusion that includes all the idealizations and corresponding caveats that are necessary to interpret our results. This latter addition will include a discussion of the perfect model framework and the representation of proxy errors in pseudoproxy experiments.

Reviewer 2 also raises the following specific points that are classified as major issues:

*...the method assumes that there is no model bias, that the model B matrix represents the true covariance structure. It is not clear how the residuals for the pseudoproxy are calculated and whether H is assumed to be known perfectly (or are the parameters of H degenerated, or is the VS-lite recalibrated somehow with the pseudoproxies?). Is R taken as diagonal? What noise model is used? And is R assumed to be known perfectly?*

Given both the perfect model and pseudoproxy frameworks, we indeed make several simplifying assumptions including some brought up here by the reviewer. B represents the true covariance structure, H is known perfectly, R is diagonal, based on white noise, and known perfectly. We will add further clarification on these specifics in Sections 2.3 and 2.4 of the revised manuscript.

*Another worry I have, in the examples given, is that the NINO index is specified with the same tree ring width series as the drought indices. Hence, the NINO index and drought are expected to be related as they are specified from the same tree ring width. Actually, this is seen in Fig. 9, which shows a stronger ENSO signal in the reconstructions using only tree ring width than in the "true" simulation. There is an element of circular reasoning here. In this context, localisation should be discussed. The author speak of localisation as an "ad hoc" method, but controlling this sort of circular reasoning would be one advantage that the authors should consider.*

Respectfully, we are not clear on the meaning of this comment. As discussed in Section 2.4 of the paper (pgs. 6-7), the state vector contains the Nino 3.4 index (along with the other variables) and we use different climate model output (temperature, precipitation, etc.) to construct the

pseudoproxies beforehand, but we are confused by the statement that the Nino index is "specified from the same tree ring width." Figure 9 shows the results from two reconstructions in which the first used only tree rings and the second used both corals and tree rings from the pseudoproxy network. A principal takeaway from Figs. 8 & 9 is that the specific proxy type used in these reconstructions doesn't matter very much for the reconstruction of ENSO (they have similar reconstruction features). As shown in Figs. 7 & 8, it is actually the reconstruction with tree rings alone that has a *weaker* ENSO signal because there is progressively less information about ENSO further afield from the central Pacific; with only remote tree-ring proxies available, adding localization would damp out even more information about ENSO and would presumably make the tree ring-only reconstructions of ENSO even worse. Moreover, as we note toward the end of Section 2.3 (pgs. 5-6), localization is actually mathematically unnecessary in the limit of very large ensemble sizes (though large ensemble sizes are a luxury we have here that isn't usually available for traditional uses of DA).

Reviewer 2 also raises some additional points that require responses beyond simple clarifications and additions (or other points addressed above) that we address below:

*P. 3, l. 29: I think the boundary conditions require some further explanations. Is it sensible not to consider boundary conditions? Or would physical consistency be violated (e.g. by using a non-volcanic background during a volcanic year?*

Several previous studies have shown that using an off-line DA approach, similar to the one that we employ, does not require boundary-condition specific priors for specific years, such as for volcanic eruption years (e.g., Steiger et al. 2014). As constructed in our manuscript, the prior contains years with volcanic eruptions that are sufficient for reconstructing volcanic eruption years.  We will clarify this point in the revised manuscript.

*P. 9, l. 4: It is not fully clear how the limitations were derived. As I understand the approach, the actual limitations are dependent on the climate conditions (couldn't the same VS-lite parameters make a tree moisture sensitive in one year but temperature sensitive in another one?).*

It is correct that the parameters of VS-lite don't determine the growth sensitivities. This point is discussed in the text indicated by the reviewer where we note that we use the growth responses (these are optional outputs of VS-lite) to compute the limitations. We will further clarify this point in the revised manuscript.

*P. 14, l. 3: In addition to El Niño, a look at the Atlantic Ocean might be interesting.*

We agree that the influence of the Atlantic Ocean would be important for real-world investigations. Nevertheless, we have not explicitly reconstructed Atlantic modes/variables in this manuscript (such as the AMOC or AMO) and the dynamic investigations that we pursue are only examples of possible analyses, none of which are meant to be exhaustive.  We therefore do not believe that an analysis of Atlantic influences on drought in the American Southwest would add significantly to the example pseudoproxy analyses.

*P. 14, l. 12-14: I find this conclusion a rather dangerous one to make in a perfect-model set-up*

The lines in the manuscript referenced here constitute an aside and are not integral to the paper. We will remove them from the revised manuscript.

---

## Author Response (AR1)

RESPONSE TO REVIEWERS

We would like to thank both reviewers for helpful and constructive comments.

**Note that line numbers that we cite refer to the track-changes manuscript in which additions are marked in blue and**

**Reviewer 1**

*My only main concern is that I do not think that the limitations of this perfect model study are made sufficiently clear. While in several places you mention that this is an "upper-bound on hydroclimate reconstruction skill" I think that this should be emphasised further, particularly in the abstract, but also while discussing the results. And it should be made clear that based on this study on its own you cannot tell if a skilful DA reconstruction is actually a possibility now, merely that it is theoretically possible. To this end I think that a section describing what uncertainty this technique includes and what it does not include would be very useful.*

This is a good idea and we have added further clarification on this point in the abstract of the revised manuscript and in new a summarizing paragraph in the description of the reconstruction experiments (p.1:8-13, p.8:10-27). We have also more clearly framed the discussion of the results in the Conclusion based on which investigated variables are unlikely to be skillfully reconstructed (the Conclusions in the original manuscript already took this perspective in the discussion of the moisture/temperature sensitivities of trees, p.19:5-7).

*I think that title should be changed from "hydroclimate extremes" to "hydroclimate variability" to reflect the more general approach of the paper.*

The first half of the paper focuses generally on hydroclimate variability, however the second half of the paper explicitly targets the extreme events of multi-year droughts. We therefore think that hydroclimate extremes is appropriate in the title.

*I found the description of the Kalman filter slightly hard to follow and think it could be improved by giving more information about what the filter is actually doing, in more accessible language so that a non-specialised audience can follow.*

We have further clarified the application of the Kalman filter in the revised manuscript (p.3:17-19).

*Do the matrices have a time dimension, or is the solution for each time step completely independent? This should be clarified.*

The solution at each time step is independent. This has been clarified at p.4:5 in the revised manuscript.

*Could multiple CESM ensembles be used for the prior information? Or does it have to be one*

*continuous simulation?*

Multiple CESM simulations could easily be used. This has been clarified at p.3:28-4:5 in the revised manuscript.

*More information should also be given about the uncertainty matrix R. Has it only diagonal terms of 0.1? Where does this come from and what uncertainty does it include and does it not include?*

More information about R and it's features have been included in the revised manuscript at the end of Section 2.3, p.6:7-7:3 and the end of Section 2.4, p.8:10-27.

*Figure 2 and others – On my print out it is quite hard to make out the position of the proxy sites in some panels.*

We have tried several different color schemes for this figure and found the one currently presented worked best. Note, however, that an additional plot of the proxy locations is included in Figure 1 without any background colors. Figure 1 therefore allows a more detailed investigation of the proxy locations, while the proxy locations in Figure 2 are provided principally to indicate how proxy skill spatially compares with broad features of the proxy network sampling. We believe Figure 2 serves this purpose, even if the fine details of the proxy sampling must be gleaned from Figure 1.

*Figure 9 - What percentiles are shown in the box plots?*

This information has been added to the caption of Figure 9 in the revised manuscript.

**Reviewer 2**

*The paper clearly gives a very optimistic upper bound. While this is certainly important, it might be more useful to the reader to give a more realistic view. Where does the method break down? What are the limitations? It is hard to judge for the reader, particularly since several optimistic assumptions are only implicit and a discussion of the limitations is missing. While I don't think that the authors should add further analyses, I definitely think they should (1) be more clear about the assumptions and (2) add a discussion on the potential limitations when applying it to the real world.*

Please see our first response to Reviewer 1 above.

*...the method assumes that there is no model bias, that the model B matrix represents the true covariance structure. It is not clear how the residuals for the pseudoproxy are calculated and whether H is assumed to be known perfectly (or are the parameters of H degenerated, or is the VS-lite recalibrated somehow with the pseudoproxies?). Is R taken as diagonal? What noise model is used? And is R assumed to be known perfectly?*

Given both the perfect model and pseudoproxy frameworks, we indeed make several simplifying assumptions including some pointed out by the reviewer in the above comment. B represents the true covariance structure, H is known perfectly, R is diagonal, based on white noise, and known perfectly. These assumptions have all been enumerated and discussed in a summarizing paragraph at p.8:10-27 in the revised manuscript.

*Another worry I have, in the examples given, is that the NINO index is specified with the same tree ring width series as the drought indices. Hence, the NINO index and drought are expected to be related as they are specified from the same tree ring width. Actually, this is seen in Fig. 9, which shows a stronger ENSO signal in the reconstructions using only tree ring width than in the "true" simulation. There is an element of circular reasoning here. In this context, localisation should be discussed. The author speak of localisation as an "ad hoc" method, but controlling this sort of circular reasoning would be one advantage that the authors should consider.*

Respectfully, we are not clear on the meaning of this comment. Figure 9 shows the results from two reconstructions in which the first used only tree rings and the second used both corals and tree rings from the pseudoproxy network. A principal takeaway from Figs. 8 & 9 is that the specific proxy type used in these reconstructions doesn't matter very much for the reconstruction of ENSO (they have similar reconstruction features). As shown in Figs. 7 & 8, it is actually the reconstruction with tree rings alone that has a *weaker* ENSO signal because there is progressively less information about ENSO further afield from the central Pacific; with only remote tree-ring proxies available, adding localization would damp out even more information about ENSO and would presumably make the tree ring-only reconstructions of ENSO even worse. Moreover, as we note toward the end of Section 2.3, localization is actually mathematically unnecessary in the limit of very large ensemble sizes (though large ensemble sizes are a luxury we have here that isn't usually available for traditional uses of DA).

*P. 3, l. 10: Give a reference for the Ensemble Square Root Filter implementation.*

Added: p.3:15.

*P. 3, l. 29: I think the boundary conditions require some further explanations. Is it sensible not to consider boundary conditions? Or would physical consistency be violated (e.g. by using a non-volcanic background during a volcanic year?*

Several previous studies have shown that using an off-line DA approach, similar to the one that we employ, does not require boundary-condition specific priors for specific years, such as for volcanic eruption years (e.g., Steiger et al. 2014). As constructed in our manuscript, the prior contains years with volcanic eruptions that are sufficient for reconstructing volcanic eruption years. This is clarified at p.3:28-4:5 in the revised manuscript.

*P. 6, l. 11: The amount of noise is really small; I am surprised by that.*

This remark is similar to the first two brought up in this review and we specifically expand the discussion of the proxy noise levels at p.6:7-7:3. Even though the measurement uncertainty here

is low, the overall pseudoproxy construction process produces pseudoproxies with signal-to-noise ratios similar real-world proxies (mean SNR values of about 0.4).

*P. 6-7: From the statement that a monthly NINO index was reconstructed I take that x_b and x_a contain both monthly and seasonal variables. How about the annual ones? Are annually and seasonal variables in the same state vector? Or are these two different experiments?*

As stated in the second sentence of section 2.4, all state variables at their relevant resolutions (monthly, seasonal, and annual) are included in x_b and x_a. Though we left out stating here that a variable with monthly resolution is also included. We have added this at p.7:6-7 of the revised manuscript.

*P. 7, l. 23: If possible within reasonable length, give equation and references for CRPSS.*

The equation is now given and discussed (references already present) on p.9:2-7 of the revised manuscript.

*P. 9, l. 4: It is not fully clear how the limitations were derived. As I understand the approach, the actual limitations are dependent on the climate conditions (couldn't the same VS-lite parameters make a tree moisture sensitive in one year but temperature sensitive in another one?).*

It is correct that the parameters of VS-lite don't determine the growth sensitivities. This point is discussed in the text indicated by the reviewer where we note that we use the growth responses (these are optional outputs of VS-lite) to compute the limitations. This is further clarified at p.9:28 of the revised manuscript.

*Fig. 2 is interesting. In the upper row (which is not really discussed in the text), the high correlations over Antarctica are striking (this could be relevant for other reconstructions). Also in the second row I find the very high skill in the tropics remarkable (the authors note it, but I think it requires more explanation). Also, it is interesting that the skill in the annual mean is smaller over the proxy sites than over the adjacent oceans. This is due to winter, where there is no skill over the proxy locations but (due to thermal inertia or other memory effects) some limited skill over the ocean. Also, in the annual mean there seem to be prominent patterns (dipoles?) in the N- and S-Pacific, pointing to very stable teleconnections within the model world.*

We agree that these are all very interesting features and we also agree with the reviewer's interpretations. We simply haven't focused on those areas with very high skill since these experiments are upper-bound skill estimates and so we think the most useful features to be those locations *without* skill because this implies that it is impossible to reconstruct these regions.

*P. 12, l. 27: I presume that the assimilation is the same as above - or not?*

We did two separate experiments, those with both corals and trees and those with just trees. This is clarified at p.14:6.

*P. 14, l. 3: In addition to El Niño, a look at the Atlantic Ocean might be interesting.*

We agree that the influence of the Atlantic Ocean would be important for real-world investigations. Nevertheless, we have not explicitly reconstructed Atlantic modes/variables in this manuscript (such as the AMOC or AMO) and the dynamic investigations that we pursue are only examples of possible analyses, none of which are meant to be exhaustive. We therefore do not believe that an analysis of Atlantic influences on drought in the American Southwest would add significantly to the example pseudoproxy analyses, and it would require significant additions to the length of the manuscript.

*P. 14, l. 12-14: I find this conclusion a rather dangerous one to make in a perfect-model set-up*

The text referenced here constitutes an aside and is not integral to the paper. We have removed this text from the revised manuscript, p.18:9-10.

[revised manuscript text omitted]